# Impact of Distribution of a Tip Sheet to Increase Early Detection and Prevention Behavior among First-Degree Relatives of Melanoma Patients: A Randomized Cluster Trial

**DOI:** 10.3390/cancers14163864

**Published:** 2022-08-10

**Authors:** Diane Marcé, Floriane Le Vilain-Abraham, Morgiane Bridou, Gaelle Quéreux, Alain Dupuy, Thierry Lesimple, Yannick Le Corre, Ewa Wierzbicka-Hainaut, Delphine Legoupil, Philippe Célérier, Hervé Maillard, Laurent Machet, Agnès Caille

**Affiliations:** 1Service de Dermatologie, Centre Hospitalier Régional Universitaire de Tours, 37044 Tours, France; 2INSERM, Université de Tours, Université de Nantes, SPHERE U1246, 37044 Tours, France; 3Laboratoire de Psychopathologie et Neuropsychologie, Université Paris 8, 93200 Paris, France; 4Dermatology Department, Nantes University Hospital, Université de Nantes, CIC 1413, Inserm UMR 1302/EMR6001 INCIT, 44000 Nantes, France; 5Service de Dermatologie, Centre Hospitalier Universitaire de Rennes, 35033 Rennes, France; 6Service d’Oncologie Médicale, Centre Anticancéreux de Rennes, 35033 Rennes, France; 7Service de Dermatologie, Centre Hospitalier Universitaire d’Angers, 49100 Angers, France; 8Service de Dermatologie, Centre Hospitalier Universitaire de Poitiers, 86000 Poitiers, France; 9Service de Dermatologie, Centre Hospitalier Universitaire de Brest, 29200 Brest, France; 10Service de Dermatologie, Centre Hospitalier de La Rochelle, 17000 La Rochelle, France; 11Service de Dermatologie, Centre Hospitalier du Mans, 72037 Le Mans, France; 12INSERM, Université de Tours, iBrain U1253, 37044 Tours, France; 13INSERM CIC1415, Centre Hospitalier Régional Universitaire de Tours, 37044 Tours, France

**Keywords:** cluster randomized trial, melanoma, prevention, screening, intervention, medical skin examination, sun-related behaviors, first-degree relatives, high-risk population, familial

## Abstract

**Simple Summary:**

The risk of melanoma is higher in first-degree relatives (FDRs, i.e., brother, sister, father, mother, or children) of a patient with melanoma than in the general population. FDRs are advised to undergo annual screening to detect any melanoma earlier and to adopt sun-protective behavior by seeking shade, wearing a hat and long-sleeved clothing, staying indoors between 12 noon and 4 p.m., and applying sunscreen with SPF > 50. We know that these general instructions, usually given orally to the patients, are inconsistently followed by FDRs. Our goal was to determine whether written support intended for FDRs of patients would improve early detection and photoprotection as compared with usual oral advice. We developed and evaluated the use of a tip sheet given by patients to their FDRs. The adherence of FDRs to early detection by medical examination and to sun protection was not improved by delivery of the tip sheet as compared with the usual oral advice.

**Abstract:**

Background: First-degree relatives (FDRs, defined as parents, children, and siblings) of melanoma patients are at a two-to-fivefold increased risk of developing melanoma themselves. FDRs are advised to perform self-skin examination (SSE) and annual medical total cutaneous examination (TCE) performed either by a dermatologist or a general practitioner, and to change their sun-related behavior. This advice is given orally to melanoma patients who are asked to relay the information to their FDRs. Objective: Our aim was to determine the impact of providing a tip sheet to melanoma patients intended to their first-degree relatives (FDRs) on early detection and sun-related behaviors in this group at increased risk of melanoma. Methods: A superiority, cluster-randomized trial was conducted at nine hospital centers. In the intervention group, dermatologists were asked to deliver to melanoma patients (index cases) the tip sheet and oral advice intended to their FDRs. The control group were asked to deliver the usual oral advice alone. The primary outcome was early detection of melanoma in FDRs with a medical TCE performed within one year after the first visit of the index case. Secondary outcomes were SSE and sun-related behaviors in FDRs. Results: A total of 48 index cases and 114 FDRS in the control group, 60 index cases and 166 FDRS in the intervention group were recruited. In the intervention group, 36.1% of FDRs performed a medical TCE as compared to 39.5% of FDRs in the control group (OR 0.9 [95% CI 0.5 to 1.5], *p* = 0.63). We did not find a between-group difference in SSE and sun-related behaviors. Conclusion: A tip sheet added to the usual oral advice did not increase medical TCE among FDRs of melanoma patients. Overall, the rate of TCE among FDRs was low. Research on other strategies is needed to increase melanoma detection in this population.

## 1. Introduction

The incidence of cutaneous melanoma (CM) still continues to increase worldwide [1] and results in 3.5 deaths per 100,000 people per year in Europe [2]. Greater tumor thickness and delayed diagnosis are associated with increased mortality. Medical total cutaneous examination (TCE) performed by a dermatologist or a general practitioner (GP), and skin self-examination (SSE) are significantly associated with thinner melanomas [3,4,5]. However, prevention and early detection on a large scale in the general population are expensive, and the potential benefit of skin cancer screening on CM mortality has not been demonstrated [6,7]. The cost of modern treatments in advanced stages has markedly increased with modern therapeutic options [8]; therefore, prevention programs may become cost-effective or even cost-saving.

First-degree relatives (FDRs; parents, siblings, children) of CM patients are at increased risk of developing CM [9,10,11]. Personalized screening of at-increased-risk individuals is recommended in many countries, including France [12]. A computer simulation study performed in the United States before the modern era of effective but costly medical treatments for advanced cancer stages found that a biennial consultation with medical TCE was cost-effective in the FDRs of CM patients [13]. Therefore, patients with newly diagnosed melanoma are advised to inform their FDRs of the increased risk to access counselling about the usefulness of protecting their skin against sun exposure and proceeding to SSE, and to consult a dermatologist or a GP annually for TCE.

The best way to deliver these messages to induce behavioral changes from FDRs remains uncertain, as well as the psychological determinants of adherence to prevention messages [14,15,16,17]. In 2012, 49% of dermatologists surveyed in the United States considered the lack of written support for FDRs a communication barrier [18]. A pilot study performed in our center suggested that a tip sheet delivered to CM patients for their FDRs which promoted sun protection behavior and medical TCE, could increase medical TCE in FDRs. We found a significant increase in FDRs’ medical TCE (50.5% without vs. 87.5% with the tip sheet, *p* = 0.006), but the TCE of FDRs was patient-reported and thus subject to misclassification and social desirability bias [19].

Here, we report the results of a cluster-randomized trial aiming to assess the impact on FDRs of the delivery to CM patients of a tip sheet in addition to usual oral advice as compared to usual oral advice.

## 2. Materials and Methods

### 2.1. Design

This was a superiority, two-parallel groups, cluster randomized controlled trial involving nine French hospital centers (six based in a university and three general) used as randomization units to limit contamination bias.

### 2.2. Setting, Participants and Data Collection

The study was conducted between January 2014 and May 2019; participants were enrolled from 13 December 2017 to 26 April 2019. Patients with CM (index cases) were informed of the study one year after the initial diagnosis of their melanoma and were asked to participate.

Figure 1 outlines the trial processes depicting order of cluster recruitment, randomization, information delivery to index cases, then identification and outcome assessments for both index cases and their FDRs; blinding status is indicated using black for complete blinding and white for no blinding [20].

During one of the first initial management visits (initial resection or wide surgery or first follow-up visit), index cases received information on the usefulness of early detection of CM and sun protection behavior for their FDRs: oral and written information in the intervention group (Appendix A) and oral information alone in the control group.

During the follow-up visit one year later, index cases were informed of the study and their non-opposition was recorded. Then, they provided contact details for their FDRs. Questionnaires and scales were completed by the index cases to assess, in particular, the actual distribution of the tip sheet to their FDRs. FDRs were then contacted by telephone. In the absence of a telephone response, FDRs were contacted by email, and in the absence of a response to this email or in the absence of an email address, by mail. After expressing their non-opposition, the data were collected from FDRs on early detection of melanoma and sun-protection behavior. In both the intervention and control groups, an information letter and non-opposition sheet, were mailed to the FDRs.

### 2.3. Selection Criteria for CM Patients and FDRs

Inclusion criteria for CM index cases were age ≥18 years, speaking French, receiving treatment for stage I or II primary cutaneous CM in one of the participating hospitals and consulting at the time of diagnosis (<3 months after the excision). Patients had to have at least one FDR. We excluded patients with mucous or ocular melanoma, who had no FDR, or who did not wish to communicate information concerning melanoma to their FDRs.

Inclusion criteria for FDRs were age ≥18 years, speaking French and first-degree related (brothers and sisters, children, father and mother) to an included index case.

### 2.4. Randomization, Intervention and Control Procedures

Hospital centers were randomly allocated to disseminate written and oral information or oral information only in a 1:1 ratio. The allocation schedule was independently created by a statistician using a computer-generated randomization list. Randomization was stratified on the type of hospital, be it university or general. This was an unblinded study.

In the intervention group, index cases received usual oral information plus a tip sheet for FDRs directly explaining to FDRs they are at increased risk of CM, should perform SSE and TCE and change their behavior regarding sun exposure by seeking shade, wearing a hat and long-sleeved clothes and using sun protection creams. The tip sheet was developed before the beginning of the trial by the medical staff and then submitted to CM patients and some of their relatives in two separate focus-group sessions lasting 2 h each to test their comprehension. The tip sheet was revised according to their feedback. The final version of the tip sheet (used in this trial) is provided in the Appendix A.

In the control group, the index cases received the usual oral information on the fact that their FDRs that they are at increased risk of CM and should perform SSE and TCE and change their behavior regarding sun exposure by seeking shade, wearing a hat and long-sleeved clothes and using sun protection creams.

The Index Cases Received Oral or Oral and Written Advice at Time T, and the Relatives were Questioned in the Month following Inclusion, at T + 12 Months.

### 2.5. Outcomes

The primary outcome was the FDRs’ participation in CM early detection by completing TCE with a dermatologist or a general practitioner performed within one year after the advice (written and oral or oral alone) had been delivered to the index cases.

Secondary outcomes were the FDRs’ planned TCE with a dermatologist or a GP with and without a scheduled appointment, FDRs’ participation in CM early detection by SSE, frequency of SSE (i.e., >one per month, one or two per year, or never) for FDRs, who declared having performed SSE and sun-protection behavior self-reported by FDRs with a questionnaire. We defined strict sun protection behavior according to the following criteria: use of high-factor sunscreen, wearing of sun-protective clothing, and/or avoidance of sun exposure.

The outcomes for FDRs were assessed during a phone call by a clinical research assistant previously trained to perform data collection. Calls were centralized in order to standardize outcome assessment. To assess the reliability of TCE self-reported by the FDRs without overburdening the trial, a random sample of 50 FDRs (25 in the control group and 25 in the intervention group) declaring to have had TCE was selected, and their physicians (dermatologist or GP) were contacted by telephone to ensure that they actually had a TCE consultation for the early detection of melanoma on the date they reported.

The outcomes for index cases were the transmission of the oral and written information to their FDRs.

### 2.6. Statistical Analysis

In our sample size calculation, we accounted for clustering at the family level rather than at the hospital level (randomization unit) because clustering was expected to be higher within families than within hospital centers. According to the data from our pilot study, we hypothesized that 60% of the FDRs in the control group will be performed a medical TCE, with an intraclass correlation coefficient at the family level of 0.40. To detect a 15% point increase in TCE, with 90% power and a 5% two-sided significance level, we needed a sample of 900 FDRs. We expected a mean number of four FDRs per CM patient and so planned to include 225 CM patients in our trial.

The baseline characteristics of index cases and FDRs are described by randomization group, with a mean (SD) or median (Q1;Q3) for continuous variables, depending on their distribution, and number (%) for categorical variables. No statistical test was performed on baseline variables.

Analysis of the primary outcome and secondary binary outcomes involved generalized estimating equation methods [21]. We used an exchangeable correlation matrix and robust variance estimators to account for clustering at the family level [22]. We used logistic regression models to estimate odds ratios (OR) with 95% confidence intervals (CIs).

In the main analysis of the primary outcome, the FDRs who performed a TCE before the initial visit of their index case and FDRs who reported a TCE but for whom the date of completion of TCE was unknown were considered not to have completed TCE within the timeframe provided in the protocol. We performed a post-hoc pragmatic sensitivity analysis considering that these FDRs had indeed a TCE (even if this was before the initial visit of the index case or at an unknown date). ICCs for the primary outcome were estimated, per randomization group, using an ANOVA estimator [23]. We estimated clustering at both hospital and family levels.

We also described the median time from the initial consultation of the index case to TCE for FDRs. No statistical test was performed on this outcome, which was not initially in the protocol. The validity of TCE self-declaration was assessed in a random sample of 50 FDRs (25 in the control group and 25 in the intervention group) who reported TCE. We describe the number of false positives (i.e., participants who reported a TCE and for whom the physician did not confirm the TCE) and the positive predictive value.

The analysis was conducted according to the intention-to-treat principle. Randomized index cases were analyzed in the group to which they were allocated. FDRs were also analyzed in the randomization group in which the index case to which they were attached was allocated. *p* < 0.05 was considered statistically significant. All analyses were performed with SAS 9.4. (SAS Institute Inc., Cary, NC, USA).

### 2.7. Ethics

Ethics review and approval were provided by the research ethics committee of CHRU Tours (no. 2016-s14), and the trial was registered at ClinicalTrials.gov (accessed on 19 June 2022) (NCT02917473).

## 3. Results

Among the nine centers, five were allocated to the intervention group (2 general hospital centers and three university hospital centers), with 75 included index cases, and four were allocated to the control group (one general hospital centers and three university hospital centers), with 66 included index cases (Figure 2). Two index cases were excluded from the intervention group because they opposed the analysis of their personal data. In the intervention group, a total of 196 FDRs were contacted, and 27 were excluded because they were not reachable or were ineligible. In the control group, five index cases were excluded (four opposed the analysis of their personal data, and one did not wish to communicate information concerning his melanoma to his relatives). In the control group, 140 FDRs were contacted, and 24 were excluded because they were not reachable or were ineligible. Finally, we included 60 index cases with 166 eligible FDRs in the intervention group and 48 index cases with 114 eligible FDRs in the control group.

### 3.1. Study Population

The characteristics of FDRs at inclusion are summarized, by the randomization group, in Table 1. The characteristics of index cases at inclusion were similar in both groups and are summarized in Appendix A.

### 3.2. Information Transmission as Reported by Index Cases

Most index cases declared having informed their FDRs about their melanoma (60/60 in the intervention group and 43/48 in the control group). In the intervention group, 59 of 60 index cases and 34 of 48 index cases in the control group declared having transmitted oral advice to FDRs. Overall, 46 of 60 index cases in the intervention group and 8 of 48 in the control arm declared having transmitted a written tip sheet to FDRs.

### 3.3. Information Transmission as Reported by FDRs

A total of 55 of 166 FDRs in the intervention group and 4 of 114 in the control group reported that they received a written tip sheet from the index case. In addition, 121 of 166 FDRs in the intervention group and 72 of 114 in the control group declared that oral advice was transmitted by the index case to FDRs.

### 3.4. Primary Outcome

In the intervention group, 60 of 166 FDRs (36.1%) reported having had a TCE by a dermatologist and/or a GP at a date after the initial consultation of the index case as compared with 45 of 114 FDRs (39.5%) in the control group (OR 0.9 [95% CI 0.5 to 1.5], *p* = 0.63) (Table 2). Sensitivity analysis accounting for all reported TCEs regardless of their date confirmed our primary results (71/166 [42.8%] in the intervention group vs. 56/114 [49.1%] in the control group) (OR 0.8 [95% CI 0.5 to 1.3], *p* = 0.39). The median time from the index case initial visit to the FDRs’ TCE was 226 days (Q1; Q3, 129; 319) in the intervention group and 250 days (129; 362) in the control group. The ICCs for reported TCE in the intervention and control groups were 0.009 and 0.06 at the hospital center level and 0.12 and 0.15 at the family level.

TCE was performed mainly by dermatologists (65/70 [92.9%] in the intervention group vs. 51/56 [91.1%] in the control group) and rarely by GPs (4/70 vs. 3/56) or by both practitioners (1/70 vs. 2/56).

Among the 127 FDRs who had a TCE with a GP or a dermatologist, 29 had a lesion removed after this consultation: 20 of 71 in the intervention group and 9 of 56 in the control group. Three melanoma cases were detected in the intervention group and one in the control group.

Among the 50 FDRs in the randomly selected subsample for validity evaluation (25 in the control group and 25 in the intervention group) declaring to have had a TCE, the self-declaration validity was assessable in 39 cases and was confirmed in all 39 cases (17 in the control group and 22 in the intervention group). Self-declaration validity was not assessable in 11 FDRs because their physicians could not be reached.

### 3.5. Secondary Outcomes

In the intervention group, 8 of 166 FDRs had a planned TCE with a scheduled appointment compared to 1 of 114 in the control group (OR 5.5 [95% CI 0.7 to 45.3], *p* = 0.39). Moreover, 36/166 FDRs in the intervention group versus 24/114 in the control group planned to perform a medical TCE without any scheduled appointment (OR 1.1 [95% CI 0.6 to 1.9], *p* = 0.88) (Table 2).

SSE was reported by 110 of 166 FDRs (66.3%) in the intervention group versus 69 of 113 (61.1%) in the control group (OR 1.3 [95% CI 0.8 to 2.2], *p* = 0.4). FDRs in the two groups did not differ according to their frequency of performing SSE (i.e., >one per month, one or two per year, or never).

A total of 121 of 166 FDRs in the intervention group and 88 of 114 in the control group reported sun-protection behavior (OR 0.8 [95% CI 0.5 to 1.4], *p* = 0.48). Components of sun-protection behaviors of the participants are summarized in Table 3. There were no participants that differed according to the randomization group.

## 4. Discussion

Our study was designed to be pragmatic and, in the case of a positive result, easily generalizable to the daily practice of dermatologists. We hypothesized that a written tip sheet given to CM patients for their FDRs, in addition to the usual oral information, would increase CM early detection in FDRs. We did not see evidence of improved engagement of FDRs in TCE recommended in the written and oral information as compared with oral information alone. Our results were consistent for the secondary outcomes, with no difference in SSE and sun protection behavior between both groups.

### 4.1. Primary and Secondary Outcomes

For the primary outcome, we found that giving a tip sheet to patients with melanoma for the FDRs did not increase the proportion of medical TCEs in FDRs as compared with the usual oral advice in the control group.

Less than 50% of FDRs in both groups actually had a TCE performed by a dermatologist or GP in the year after the removal of the melanoma in their relative. The low rate of TCE observed in both groups of FDRs is consistent with a study conducted from 2001 to 2003 in the United States [24]. The authors found that 45% of FDRs never had a TCE and that 13.4% had not had a TCE within the past three years. In the same study, 28% of FDRs in the intervention group versus 39% in the control group had not performed SSE in the past year. This proportion is close to 34% in the intervention group in our study. The frequency of SSE among FDRs varies from 30.8% to 71.6%, depending on the study [24,25].

Three previously published randomized trials conducted in the United States found improvements in either TCE or SSE with a tailored intervention directed to FDRs as compared to usual care [26,27,28]. The interventions assessed in these trials all involved repeated messages dedicated directly to FDRs.

Indeed, our intervention was more pragmatic than those previously evaluated because it consists of a unique message and relies on patients rather than FDRs. However, because our intervention did not directly target the FDRs, to be successful, it required that first the patient transmit the tip sheet to their FDRs and then the FDRs adhere to the screening and sun protection behaviors. Thus, some elements may explain the failure of our intervention.

First, the rate of FDRs who declared having received the tip sheet was low: 33.1% in the intervention group versus 3.5% in the control group. These results are also in favor of a small between-group contamination or a desirability bias from FDRs in the control group. This low rate of transmission of the tip sheet can be explained by the evolution of what primarily worries the most patients and their families after the announcement of the diagnosis for the patients: at first, family conversations typically focus on the patient, including the diagnosis and the treatment, and after the acute treatment phase, conversations about family risk and prevention predominate [29]. Hence, the distribution of the tip sheet close to the diagnosis stage may have occurred too early. During this second step, melanoma patients may not inform their FDRs because of the perception of low or high risk of occurrence of melanoma in their FDRs according to skin color, ability to sunburn and age [30]. Hence, tailored and repeated interventions should increase the rate of information transmitted to FDRs. During this step, the tip sheet can be lost, forgotten, or difficult to transmit because relatives and index cases do not live in the same city, for example. Thus, the paper support may not be the most suitable support. In addition, FDRs may directly search the Internet for information and find the tip sheet of little use. Indeed, in our study, 72.5% and 64.3% of FDRs in the intervention and control groups, respectively, declared having frequently sought information on the Internet. Moreover, the information on skin cancers, sun protection behaviors, and family risk of cancers has been widely diffused in the media in France, with a “Skin Cancer Day” conducted every year. Accordingly, many more patients and their families are now better informed compared to the early 2000s.

Second, knowing the risks of skin cancer (or other cancers) does not imply engagement in a screening program and reduction or avoidance of risk factors due to an addiction to known carcinogens (e.g., tobacco use or sun exposure) [31].

### 4.2. Strengths and Limitations

The main strength of our study is its pragmatic design. Because dermatologists have underlined the absence of standardized guidelines and a lack of written material as moderate barriers to melanoma communication, we aimed to design a simple and low-cost intervention to improve sun protection and early detection of melanoma in FDRs. Moreover, the tip sheet was established with the help of two focus groups conducted by a psychologist and a dermatologist with both patients and FDRs. Cluster randomization is also a strength of the trial because it allows all the members of a given family to be in the same randomization group. It could have created a family effect on screening behavior. Another strength of the study is that the main outcome was not only self-declared because we verified that a TCE was indeed performed by calling dermatologists and GPs for a random sample of 50 FDRs who declared TCE.

The main limitation is the lack of power. We planned to include 900 FDRs and finally included 279. Indeed, we overestimated the expected number of FDRs: we estimated the number of FDRs per CM patient to be four, whereas patients reported only 2.6 FDRs, on average, which is the same as reported by Manne et al. [27]. Because of its design, it was not possible to extend the inclusion period of the trial to increase the sample size. Another limitation is that the advice sheet was based on text only, without any visual support such as drawings or pictures [32]. Moreover, after the tip sheet was established with the help of two focus groups, the relevance of the information was not evaluated in individual cognitive interviews with FDRs or further qualitative assessment with Likert scales to assess the value of the content of the tip sheet. Finally, the perceived severity of skin cancer/melanoma assessed among the FDRs as well other psychological factors and the health professional’s ability to provide clear explanations [33] may influence the impact of the intervention.

## 5. Conclusions

Our trial did not provide evidence that a tip sheet added to the usual oral information delivered to melanoma patients for their FDRs improved screening or sun-protection behaviors. Such behavior modification likely requires direct and more intensive personalized and repeated interventions.

## Figures and Tables

**Figure 1 cancers-14-03864-f001:**
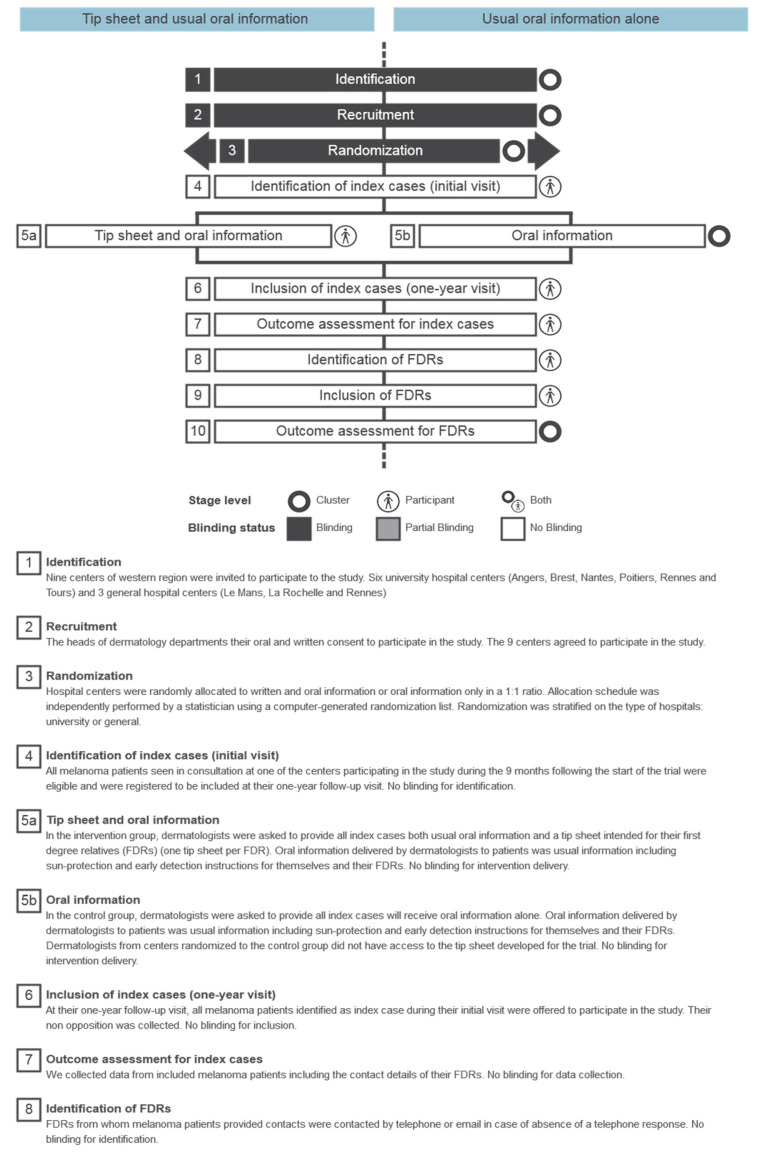
Timeline cluster diagram for the trial.

**Figure 2 cancers-14-03864-f002:**
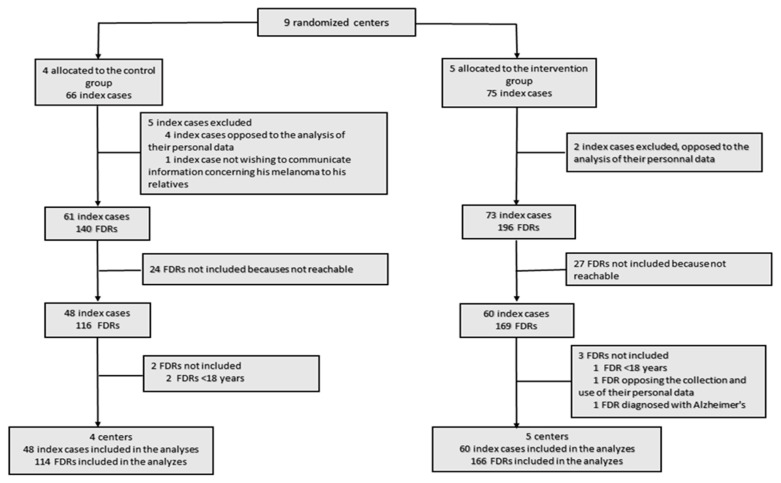
The flow of clusters, index cases and first-degree relatives (FDRs) in the study.

**Table 1 cancers-14-03864-t001:** Socio-demographic characteristics and melanoma risk factors for first-degree relative (FDRs) at inclusion by randomization group.

	Intervention Group Written and Oral Information n = 166	Control Group Oral Information Alone n = 114
FDR characteristics		
Age, n_i_ = 165, n_c_ = 114	52 (17)	50 (18)
Men, n_i_ = 166, n_c_ = 114	75 (45.2)	45 (39.5)
Level of education, n_i_ = 166, n_c_ = 114		
Secondary school	24 (14.5)	15 (13.2)
Certificate of professional competence/Professional study certificate	34 (20.5)	26 (22.8)
High school diploma	43 (25.9)	38 (33.3)
Bachelor’s degree	32 (19.3)	16 (14.0)
Master’s degree	32 (19.3)	18 (15.8)
Other	1 (0.6)	1 (0.9)
Occupational status, n_i_ = 166, n_c_ = 114		
Full-time professional	95 (57.2)	59 (51.8)
Part-time professional	10 (6.0)	7 (6.1)
No activity	1 (0.6)	0 (0.0)
Work time accident/occupational disease	2 (1.2)	0 (0.0)
Student	6 (3.6)	7 (6.1)
Disability	1 (0.6)	0 (0.0)
Unemployment	0 (0.0)	3 (2.6)
Retired	49 (29.5)	36 (31.6)
Other	2 (1.2)	2 (1.8)
Business, n_i_ = 165, n_c_ = 113		
Farmer	2 (1.2)	0 (0.0)
Artisan, trader, head of enterprise	10 (6.1)	6 (5.3)
Executive, higher intellectual profession	25 (15.2)	17 (15.0)
Intermediate occupation	11 (6.7)	2 (1.8)
Employed	58 (35.2)	38 (33.6)
Worker	1 (0.6)	3 (2.7)
Other	58 (35.2)	47 (41.6)
Relationship status to index case, n_i_ = 165, n_c_ = 114		
Father or mother	20 (12.1)	17 (14.9)
Brother or sister	69 (41.8)	41 (36.0)
Son or daughter	76 (46.1)	56 (49.1)
Family situation, n_i_ = 165, n_c_ = 114		
Married	113 (68.5)	76 (66.7)
Single	36 (21.8)	19 (16.7)
Widow (er)	6 (3.6)	5 (4.4)
Separated/divorced	10 (6.1)	14 (12.3)
FDRs with children, n_i_ =165, n_c_ = 114	117 (70.9)	80 (70.2)
Number of children, n_i_ = 117, n_c_ = 80	2 (2; 3)	2 (2; 3)
History of melanoma, n_i_ = 161, n_c_ = 112	6 (3.7)	3 (2.6)
History of another cancer, n_i_ = 163, n_c_ = 113	12 (7.4)	10 (8.8)
Skin cancer *, n_i_ = 12, n_c_ = 10	5 (41.7)	6 (60.0)
Other type of cancer *, n_i_ = 12, n_c_ = 10	7 (58.3)	5 (50.0)

Data are n (%) or mean (SD) or median [Q1; Q3]; * a patient could have several histories of cancer; n_i_, intervention group; n_c_, control group.

**Table 2 cancers-14-03864-t002:** First-degree relatives’ total cutaneous examination (TCE) performed by a dermatologist and/or a general practitioner for the early detection of melanoma by randomized group made an appointment or planning to make an appointment for an early detection examination.

	Intervention Group Written and Oral Information n = 166	Control Group Oral Information n = 114	OR [95% CI]	*p*-Value
Primary outcome: TCE performed strictly within the period of the study *	60 (36.1%)	45 (39.5%)	0.9 [0.5 to 1.5]	0.63
Sensitivity analysis of the primary outcome: all TCEs performed, including TCE performed just before the inclusion period or with a missing date of appointment	71 (42.8%)	56 (49.1%)	0.8 [0.5 to 1.3]	0.39
Secondary outcomes				
TCE planned with a scheduled appointment but not yet completed	8 (4.8%)	1 (0.9%)	5.5 [0.7 to 45.3]	0.11
TCE planned with no scheduled appointment yet	36 (21.7%)	24 (21.1%)	1.1 [0.6 to 1.9]	0.88

OR—odds ratio; 95% CI, 95% confidence interval; * In this analysis, patients who had a screening examination at an unknown date or before the initial visit of their index case were considered as not having had a TCE (i.e., as a “failure” for the primary outcome).

**Table 3 cancers-14-03864-t003:** Sun protection behaviors of first-degree relatives.

	Intervention Group Written and Oral Information n = 166	Control Group Oral Information n = 114	OR [95% CI]	p-Value
Use of sun protection	121 (72.9%)	88 (77.2%)	0.8 [0.5 to 1.4]	0.48
Avoidance of sun exposure	39 (23.5%)	22 (19.3%)	1.4 [0.7 to 2.7]	0.40
Wearing protective clothes	70 (57.9%)	38 (43.2%)	1.5 [0.9 to 3.3]	0.13
Use of high index sunscreen	105 (63.3%)	70 (61.4%)	1.1 [0.6 to 1.9]	0.73

OR—odds ratio; 95% CI—95% confidence interval.

## Data Availability

All data used in the study are available from the corresponding author upon request.

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
