# Peer review of "Impact of Distribution of a Tip Sheet to Increase Early Detection and Prevention Behavior among First-Degree Relatives of Melanoma Patients: A Randomized Cluster Trial"

_cancers, 2022, doi:10.3390/cancers14163864_

Round 1
Reviewer 1 Report
This well designed study sought to determine whether written support via a tip sheet for FDRs of patients with melanoma would improve early detection and photoprotection as compared with usual oral advice. The intervention group received oral advice and the tip sheet and the control received oral advice alone. The superiority, cluster-randomized trial found that adherence of FDRs was not improved by the tip sheet in comparison with the usual oral advice for either the primary endpoint of the annual medical total cutaneous exam or secondary outcomes of SSE and sun-related behaviors. This pragmatic study provided invaluable results despite it being a negative study. Despite the lack of power, we know that relying on melanoma patients to provide information to FDRs is not a reliable method of empowering and engaging FDRs in skin cancer early detection and prevention.
Major concern
The relationship status of the FDR to the melanoma patient needs to be stated. By the age of the index cases (melanoma patients) (63,61 years in Table S1) and the age of the FDR (52, 50 in Table 1), it seems that the FDRs are siblings. The “closeness” of the relationship between the patient and the selected FDR may influence the transmission of the tip sheet and if oral communication was provided by the melanoma patient to the FDR. This point may be added to the discussion paragraph lines 315-333.
This paragraph is an excellent discussion of the difficulty in sorting out the many factors that influence the transmission of information from the melanoma patient to a FDR.
Minor consideration:
1. The provided tip sheet in the supplementary file (in French) seems to be all text. The work of others has shown that color images and diagrams aid comprehension and perhaps help to motivate performance of TCE and SSE. The research group performed 2 separate focus-group sessions lasting 2 hr each to develop it. It would have been helpful to perform individual cognitive interviews with FDRs and further qualitative assessment with Likert scales about the value of the content. This type of interview would have explored the attitudes of FDRs about the relevance of the information to them.
2. Lines 233 -234
Overall, 46 of 60 index cases in the intervention group 233 and 8 of 48 in the control arm declared having transmitted a written tip sheet to FDRs.
This should be: “declared having transmitted a written tip sheet to FDRs or discussed oral instructions”.
Lines 236-238 Also have the confusion a4 of114 in the control group reported receiving a written tip sheet. If this is true, it raises concern about contamination of the control group.
3. Citation in line 299 should be 20 not 19
Author Response
Reviewer 1. This well designed study sought to determine whether written support via a tip sheet for FDRs of patients with melanoma would improve early detection and photoprotection as compared with usual oral advice. The intervention group received oral advice and the tip sheet and the control received oral advice alone. The superiority, cluster-randomized trial found that adherence of FDRs was not improved by the tip sheet in comparison with the usual oral advice for either the primary endpoint of the annual medical total cutaneous exam or secondary outcomes of SSE and sun-related behaviors. This pragmatic study provided invaluable results despite it being a negative study. Despite the lack of power, we know that relying on melanoma patients to provide information to FDRs is not a reliable method of empowering and engaging FDRs in skin cancer early detection and prevention.
We thank the reviewer for this comment
Major concern
The relationship status of the FDR to the melanoma patient needs to be stated. By the age of the index cases (melanoma patients) (63,61 years in Table S1) and the age of the FDR (52, 50 in Table 1), it seems that the FDRs are siblings. The “closeness” of the relationship between the patient and the selected FDR may influence the transmission of the tip sheet and if oral communication was provided by the melanoma patient to the FDR. This point may be added to the discussion paragraph lines 315-333.
In the revised version of the manuscript, relationship status of the FDR has been added to Table 1 characteristics. Indeed, most of them were son or daughter of the index case; their mean age was 40.3 years and 36.6 years in the intervention and control group respectively. For brothers and sisters, mean age was 59.9 years and 60.1 years, respectively and for mothers and fathers of the index case it was 70.0 and 71.0, respectively. Because there was no significant difference between intervention and control group, we do not feel necessary to discuss it.
This paragraph is an excellent discussion of the difficulty in sorting out the many factors that influence the transmission of information from the melanoma patient to a FDR.
Minor consideration:
1-The provided tip sheet in the supplementary file (in French) seems to be all text. The work of others has shown that color images and diagrams aid comprehension and perhaps help to motivate performance of TCE and SSE.
We agree with the reviewer. This has been added in the limitation section “Another limitation is that the advice sheet was based on text only, without any visual support such as drawing or pictures [32]. “ with an additional reference (McWhirter, J.E.; Hoffman-Goetz, L. Visual images for patient skin self-examination and melanoma detection: a systematic review of published studies. J Am Acad Dermatol. 2013, 69:47-55.)
2-The research group performed 2 separate focus-group sessions lasting 2 hr each to develop it. It would have been helpful to perform individual cognitive interviews with FDRs and further qualitative assessment with Likert scales about the value of the content. This type of interview would have explored the attitudes of FDRs about the relevance of the information to them.
We agree with the reviewer. This has been added in the limitation section. “Moreover, after the tip sheet was established with the help of 2 focus groups , the relevance of information was not evaluated with individual cognitive interviews with FDRs and further qualitative assessment with Likert scales to assess the value of the content of the tip sheet. »
3- Lines 233 -234
Overall, 46 of 60 index cases in the intervention group and 8 of 48 in the control arm declared having transmitted a written tip sheet to FDRs.
This should be: “declared having transmitted a written tip sheet to FDRs or discussed oral instructions”.
Yes this was unexpected but 8 index cases in the control arm declared having received and further transmitted a written tip sheet.
4- Lines 236-238 Also have the confusion a 4 of 114 in the control group reported receiving a written tip sheet. If this is true, it raises concern about contamination of the control group.
Indeed, this is an unexpected results but some index cases in the control group actually declared they had transmitted a written tip sheet and some FDRs in the control group reported they had received a written tip sheet. This could be due to either social desirability bias, to memory impairment or confusion with another written form. We don’t feel contamination between group was possible, and the randomization was done at the level of the hospital to avoid contamination.
- Citation in line 299 should be 20 not 19.
Done, and now n°23 because 3 references were added in the methods section

Reviewer 2 Report
Thank you for this well-written manuscript presenting an interesting study. I have minor comments:
1. Line 75, there seems to be an error in this sentence (see the ?)
2. More references are required for the analysis methods, particularly in line 177-179
3. Could the tip sheet be made available in English as well?
4. I may have missd it, but I'm not clear on when and how the behaviours of the FDRs are captured? It would be good to clarify this in the Methods
5. Line 318, this sentence could be better worded as it's not clear to me what the temporality is referring to.
6. Other than the tip sheet, were the index cases provided any guidance on how to discuss the prevention and early detection advice? Does the tip sheet provide any details about the FDR personal skin cancer risk? Was perceived severity of skin cancer/melanoma measured amongst the FDRs? This may have been another factor influencing the limited impact of the intervention.
Author Response
Reviewer 2. Thank you for this well-written manuscript presenting an interesting study. I have minor comments:
We thank you for your comments.
- Line 75, there seems to be an error in this sentence (see the ?)
? deleted
- More references are required for the analysis methods, particularly in line 177-179
3 references were added :
- Zeger, S.L.; Liang, K.Y. Longitudinal data analysis for discrete and continuous outcomes. Biometrics. 1986, 42,124-130.
- Eldridge, S.; Kerry, S.M. A practical guide to cluster randomised trials in health services research. John Wiley & Son: Chichester, United Kingdom, 2011.
- Ridout, M.S.; Demetrio, C.G.B.; Firth, D. Estimating Intraclass Correlation for Binary Data. Biometrics. 1999, 55, 137-148.
- Could the tip sheet be made available in English as well?
Done
- I may have missd it, but I'm not clear on when and how the behaviours of the FDRs are captured? It would be good to clarify this in the Methods
We added in the methods section paragraph 2-4:
« The index cases received oral or oral and written advice at a time T, and the relatives were questioned in the month following inclusion, carried out at T +12 months. »
- Line 318, this sentence could be better worded as it's not clear to me what the temporality is referring to.
We have modified this sentence as follows: “This low rate of transmission of the tip sheet can be explained by the evolution of what primarily worry the most patients and their family after the announcement of the diagnosis for the patients: at first, family conversations typically focus on the patient, including the diagnosis and the treatment, and after the resolution of this acute treatment phase, conversations about family risk and prevention predominate [29]. Distribution of the tip sheet close to the diagnosis stage may thus have occurred too early.”
- Other than the tip sheet, were the index cases provided any guidance on how to discuss the prevention and early detection advice?
No specific guidance, except the tip sheet, was provided to index cases
- Does the tip sheet provide any details about the FDR personal skin cancer risk?
The tip sheet has been translated. No detailed information was present in the tip sheet. FDRs were informed about an increased risk of developping a melanoma.
- Was perceived severity of skin cancer/melanoma measured amongst the FDRs? This may have been another factor influencing the limited impact of the intervention.
We agree. The following sentence was added in the limitation section. Finally, perceived severity of skin cancer/melanoma measured amongst the FDRs as other psychological factors may influence the impact of the intervention.
